# Minimally Invasive Preclinical Monitoring of the Peritoneal Cavity Tumor Microenvironment

**DOI:** 10.3390/cancers14071775

**Published:** 2022-03-31

**Authors:** Zachary Spencer Dunn, Yan-Ruide Li, Yanqi Yu, Derek Lee, Alicia Gibbons, James Joon Kim, Tian Yang Zhou, Mulin Li, Mya Nguyen, Xinjian Cen, Yang Zhou, Pin Wang, Lili Yang

**Affiliations:** 1Mork Family Department of Chemical Engineering and Materials Science, University of Southern California, Los Angeles, CA 90089, USA; zacharsd@usc.edu (Z.S.D.); pinwang@usc.edu (P.W.); 2Department of Microbiology, Immunology & Molecular Genetics, University of California, Los Angeles, CA 90095, USA; charlie.li@ucla.edu (Y.-R.L.); yu63@g.ucla.edu (Y.Y.); ylee932@ucla.edu (D.L.); acgibbons@g.ucla.edu (A.G.); jooniekim1@ucla.edu (J.J.K.); zhou603@usc.edu (T.Y.Z.); mulin@g.ucla.edu (M.L.); mnguyen1198@g.ucla.edu (M.N.); xicen@g.ucla.edu (X.C.); zzydcat@g.ucla.edu (Y.Z.); 3Eli and Edythe Broad Center of Regenerative Medicine and Stem Cell Research, University of California, Los Angeles, CA 90095, USA; 4Jonsson Comprehensive Cancer Center, David Geffen School of Medicine, University of California, Los Angeles, CA 90095, USA; 5Molecular Biology Institute, University of California, Los Angeles, CA 90095, USA

**Keywords:** immune cell monitoring, intraperitoneal models, cell therapy, preclinical ovarian cancer

## Abstract

**Simple Summary:**

The mammalian peritoneal cavity harbors the liver, spleen, most of the gastrointestinal tract and other viscera, and is a dynamic microenvironment involved in numerous biological and pathological processes. Here, we present a simple, novel method termed in vivo intraperitoneal lavage (IVIPL) for the minimally invasive monitoring of cells in the peritoneal cavity of mice. IVIPL was used to study changes in the cellular composition of intraperitoneal tumor microenvironments in a syngeneic model of ovarian cancer and a xenograft CAR-T cell-treated ovarian cancer model, validating the application of IVIPL to monitor preclinical intraperitoneal cellular evolution in real time.

**Abstract:**

Intraperitoneal (i.p.) experimental models in mice can recapitulate the process of i.p. dissemination in abdominal cancers and may help uncover critical information about future successful clinical treatments. i.p. cellular composition is studied in preclinical models addressing a wide spectrum of other pathophysiological states such as liver cirrhosis, infectious disease, autoimmunity, and aging. The peritoneal cavity is a multifaceted microenvironment that contains various immune cell populations, including T, B, NK, and various myeloid cells, such as macrophages. Analysis of the peritoneal cavity is often obtained by euthanizing mice and performing terminal peritoneal lavage. This procedure inhibits continuous monitoring of the peritoneal cavity in a single mouse and necessitates the usage of more mice to assess the cavity at multiple timepoints, increasing the cost, time, and variability of i.p. studies. Here, we present a simple, novel method termed in vivo intraperitoneal lavage (IVIPL) for the minimally invasive monitoring of cells in the peritoneal cavity of mice. In this proof-of-concept, IVIPL provided real-time insights into the i.p. tumor microenvironment for the development and study of ovarian cancer therapies. Specifically, we studied CAR-T cell therapy in a human high-grade serous ovarian cancer (HGSOC) xenograft mouse model, and we studied the immune composition of the i.p. tumor microenvironment (TME) in a mouse HGSOC syngeneic model.

## 1. Introduction

The peritoneal cavity is a membrane-bound, fluid-filled space within the abdomen that contains the intestines, stomach, and liver [1]. Peritoneal fluid is normally small in volume and harbors diverse cell populations, including neutrophilic leukocytes, macrophages, and lymphocytes [2]. The peritoneal cavity and the pathological accumulation of peritoneal fluid, known as ascites, have been implicated in the progression of a variety of diseases, including liver cirrhosis, heart or kidney failure, infection, and cancer [3].

Gastrointestinal and gynecological cancers, such as ovarian, colorectal, and stomach carcinomas, often present with late-stage disease and are recalcitrant to traditional therapies as well as novel immunotherapies, resulting in high patient mortality [4,5]. At advanced stages, 70% of ovarian and 25% of colon cancer patients exhibit peritoneal carcinomatosis, a form of cancer progression associated with median survival rates of less than 6 months [6]. Malignant ascites is present in more than one-third of ovarian cancer patients at initial diagnosis and almost all cases of relapse, and can facilitate metastasis and contribute to chemoresistance as well as patient morbidity and mortality [7,8]. New therapies and strategies to combat ovarian cancers and peritoneal metastasis are urgently needed, and preclinical mouse models may provide essential insights into the development of such advancements.

Cell therapies are rapidly becoming a new pillar of cancer treatment. Expansion and infusion of tumor-infiltrating lymphocytes resulted in notable clinical efficacy in melanoma patients [9], and chimeric antigen receptor T (CAR-T) cell therapy, in which a patient’s T cells are genetically engineered to express a synthetic CAR linking a cancer-targeting domain to T cell stimulatory domains, has transformed the treatment of hematological malignancies, as evidenced by five FDA-approved CAR-T cell products for B cell cancers [10,11]. CAR-T cells have yet to achieve widespread success in combating solid tumors, including ovarian cancers [12]. Solid tumors present additional hurdles to the antitumor efficacy of CAR-T cells, including barriers to tumor infiltration, the conception of an immunosuppressive tumor microenvironment (TME), and a lack of tumor-specific antigens/heterologous antigen expression [13]. The past decade has witnessed a particular focus on the various cellular populations, soluble factors, and structural proteins that foster an immunosuppressive TME [14].

The peritoneal TME contains cancer cells, stromal cells, and immune cells, and the interactions between them influence tumor progression and immune escape and may represent specific targets for immunotherapy [15]. For example, chemotherapy skews the tumor associated macrophage (TAM) population in HSGOC toward an M1 antitumor phenotype (rather than protumor M2 phenotype) that aids adaptive immunity, and macrophage depletion via CSF1R inhibitors after chemotherapy treatment reduced adaptive antitumor immune responses and significantly decreased disease-free and overall survival in mice [16]. Therapies that enhance or sustain antitumor macrophages during chemotherapy-induced remission can potentially delay ovarian cancer relapse [16]. In a 2020 prospective observational study of ovarian cancer patients, there was a significant positive association between the M1/M2 ratio and an improved OS, PFS, and platinum-free interval (PFI), both in the entire population and in patients stratified according to tumor type and initial surgery [17]. Macrophages represent one promising cellular target in the TME—fibroblasts, endothelial cells, and other cell populations can also influence malignant ascites progression [15] and each can be targeted by various modalities [18,19]. A better understanding of TME dynamics, which vary between cancer types and subtypes and in response to therapies, may lead to optimized combinatory treatment regimens that outmaneuver TME evolution.

A common and straightforward method for studying cancer progression and preclinically assessing ovarian cancer treatments, including CAR-T cells, is to implant human or mouse cancer cells into mice by i.p. injection. Traditionally, analysis of the peritoneal cavity requires euthanasia of the mouse and terminal peritoneal lavage [20]. This necessitates the use of more mice, reagents, and time to assess the evolution of the i.p. microenvironment and may prevent i.p. cellular analysis or limit it to a single time point. We hypothesized that i.p. injection of saline into live mice followed by peritoneal fluid aspiration would provide a minimally invasive method for monitoring immune cell composition of the peritoneal cavity. In this proof-of-concept, we employed our new method, termed in vivo intraperitoneal lavage (IVIPL), for the study of ovarian cancer in immunocompromised and syngeneic mouse models. We assessed the in vivo dynamics of mesothelin-targeted CAR-T cells in an OVCAR8 i.p. model of HGSOC. Furthermore, using IVIPL, we characterized cancer cell and mouse myeloid cell phenotypic alterations in response to CAR-T cell treatment, which may indicate potential targets for combination therapies. We also validated IVIPL in a syngeneic model of HGSOC by studying i.p. cell populations during the tumorigenesis of ID8 cancer in B6 mice. As i.p. preclinical models are used to study a wide range of pathologies, we hope that IVIPL can provide increased access to time-course insights into i.p. cellular evolution.

## 2. Materials and Methods

### 2.1. Mice and Cells

Six- to ten-week-old female mice were used for all experiments. NOD.Cg-PrkdcSCIDIl2rgtm1Wjl/SzJ (NOD/SCID/IL-2Rγ-/-, NSG) mice were maintained in the animal facilities of the University of California, Los Angeles (UCLA). C57BL/6J (B6) mice were purchased from the Jackson Laboratory. All animal experiments were approved by the Institutional Animal Care and Use Committee of UCLA. Human embryonic kidney 293T cells and high-grade serous ovarian cancer cell lines OVCAR3 and OVCAR8 were purchased from the American Type Culture Collection (ATCC), and ID8 murine ovarian cancer cell line was purchased from Millipore. The 293T cells, and OVCAR3 and OVCAR8 cell lines, were cultured in DMEM supplemented with 10% (*v*/*v*) FBS and 1% (*v*/*v*) penicillin/streptomycin/glutamine (D10 medium), and ID8 was cultured in D4 medium (4% FBS). To make stable tumor cell lines expressing firefly luciferase and enhanced green fluorescence protein (Fluc-EGFP) dual-reporters, the parental tumor cell lines were transduced with lentiviral vectors encoding the intended genes [21]. Cells were subjected to flow cytometry sorting to isolate gene-engineered cells for making stable cell lines 72 h post lentivector transduction. Two stable tumor cell lines were generated for this study, including OVCAR3-FG and OVCAR8-FG. Healthy donor human peripheral blood mononuclear cells (PBMCs) were obtained from UCLA/CFAR Virology Core Laboratory and cultured in RPMI 1640 supplemented with FBS (10% *v*/*v*), P/S/G (1% *v*/*v*), MEM NEAA (1% vol/vol), HEPES (10 mM), Sodium Pyruvate (1 mM), b-ME (50 mM), and Normocin (100 mg/mL) (C10 medium). To produce CAR-T cells, PBMCs were stimulated with CD3/CD28 T-activator beads (ThermoFisher Scientific, Waltham, MA, USA) in the presence of recombinant human IL-2 (30 ng/mL). On day 2, cells were spin-infected with frozen-thawed mesothelin-targeting chimeric antigen receptor retroviral vector (RV/MCAR) supernatants supplemented with polybrene (10 mg/mL, Sigma-Aldrich, St. Louis, MO, USA) at 660 g at 30 °C for 90 min. The resulting MCAR-T cells were expanded for another 7–10 days, and then were cryopreserved for future use. CAR-T cells were thawed and recovered for 24 h in C10 supplemented with IL-7 (10 ng/mL) and IL-15 (10 ng/mL) prior to in vivo application. Vsv-g-pseudotyped RV/MCAR retroviruses were generated by transfecting HEK293T cells following a standard calcium precipitation protocol.

### 2.2. Mouse Tumor Models

In studies using OVCAR8FG cells, NSG mice were inoculated i.p. with 1 × 10^5^ OVCAR8FG cells (Day 0). On day 4, the tumor-bearing experimental mice received intraperitoneal (i.p.) injection of vehicle (PBS), or 5 × 10^6^ MCAR-T cells. In studies using OVCAR3FG cells, NSG mice were inoculated with 3 × 10^6^ OVCAR3FG cells (Day 0). On day 20, the tumor-bearing experimental mice received i.p. injection of vehicle (PBS), or 5 × 10^6^ MCAR-T cells. Tumor loads and mice weights were monitored by measuring total body luminescence using BLI at least two times per week using a Spectral Advanced Molecular Imaging (AMI) HTX imaging system (Spectral instrument Imaging, Tucson, AZ, USA). Live animal imaging was acquired 5 min after intraperitoneal (i.p.) injection of D-Luciferin (1 mg per mouse). Imaging results were analyzed using AURA imaging software (Spectral Instrument Imaging). In studies using ID8 cells, B6 mice were inoculated i.p. with 5 × 10^6^ ID8 cells. Mice were weighed three times per week and checked daily for clinical signs of ascites (swollen bellies) and evidence of toxicity, such as hunched posture, weight loss, change in mobility, change in behavior, and failure to eat or drink. Following institutional guidelines, mice were euthanized when they developed ascites and had a weight increase >30% of their original weight on day 1.

For all the mouse tumor models, IVIPL (described below) was performed at various time points to analyze cell populations in the intraperitoneal cavity. For some experiments, retro-orbital bleeding [22] was performed to analyze cell populations in the peripheral blood. For some experiments, following euthanasia, terminal peritoneal lavage was performed [20].

### 2.3. In Vivo Intraperitoneal Lavage (IVIPL)

IVIPL is a simple, minimally invasive technique to monitor cells in the i.p. cavity of mice. Throughout the IVIPL procedure, the mice remain under isoflurane anesthesia to limit the suffering of the animals. First, the mouse is anesthetized using isoflurane induction. A volume of 500 uL of 1× phosphate buffered saline (PBS) is injected i.p. Then, the abdomen of the mouse is gently kneaded, and the mouse is returned to the isoflurane chamber. The mouse is then removed from the chamber shortly thereafter and an empty 27 g needle is inserted for i.p. injection to avoid organs, after which the mouse is rotated to allow peritoneal fluid accumulation in the lower abdomen and 50 uL of fluid is aspirated. The aspirate is transferred to a collection tube containing a small volume (200 uL) pre-chilled C10 and is ready for downstream processing. After performing IVIPL, we observe the mice for a few minutes for any adverse events (hunching, excessive grooming, bleeding, etc.) and 24 h later for signs of abdominal swelling or generalized illness (peritonitis). If any are seen, we will euthanize the mouse or contact a DLAM veterinarian. IVIPL can be performed at a maximum frequency of once per day and no more than three times per week for the duration of the experiment.

### 2.4. Antibodies and Flow Cytometry

For flow cytometry analysis, all tissues were processed into mononuclear cells (MNCs) and lysed of red blood cells (RBCs) with Tris-buffered ammonium chloride (TAC) buffer, following a standard protocol (Cold Spring Harbor Protocols). All flow cytometry stains were performed in PBS for 15 min at 4 °C. The samples were stained with Fixable Viability Dye eFluor506 (e506) mixed with Mouse Fc Block (anti-mouse CD16/32) prior to antibody staining. Antibody staining was performed at a dilution according to the manufacturer’s instructions. Fluorochrome-conjugated isotypes and antibodies specific for human CD45 (Clone H130), CD3 (Clone I26), CD4 (Clone OKT4), CD8 (Clone SK1), CD279 (PD-1) (Clone A17188B), CD336 (TIM-3) (Clone A18087E), CD274 (PD-L1) (Clone 29E.2A3), and mouse CD45 (Clone 30-F11), CD45.2 (Clone 104), TCRβ (Clone H57-597), CD4 (Clone RM4-5), CD8 (Clone 53-6.7), CD279 (PD1) (Clone RMP1-30), CD366 (TIM-3) (Clone RMT3-23), CD25 (Clone PC61), Gr-1 (Clone RB6-8C5), CD11b (Clone M1/70), CD206 (Clone C068C2), F4/80 (Clone BM8), CD86 (Clone GL-1), and CD11c (Clone N418) were purchased from BioLegend (San Diego, CA, USA). Fluorochrome-conjugated isotype and antibody specific for human mesothelin (Clone 420411) were purchased from R&D Systems. Fluorochrome-conjugated antibody specific for mouse IL-2 (Clone JES6-5H4) was purchased from BDBiosciences. Fluorochrome-conjugated antibodies specific for mouse NK1.1 (Clone PK136) and FOXP3 (Clone FJK-16s) were purchased from eBioscience (San Diego, CA, USA). Biotinylated mesothelin and fluorochrome-conjugated streptavidin were purchased from human BioLegend. Mouse Fc Block (anti-mouse CD16/32) was purchased from BD Biosciences. Fixable Viability Dye e506 was purchased from Affymetrix eBioscience. Stained cells were analyzed using a MACSQuant Analyzer 10 flow cytometer (Miltenyi Biotech, Bergisch Gladbach, Germany). FlowJo software was utilized to analyze the data.

### 2.5. Statistical Analysis

GraphPad Prism 8.0.1 (Graphpad Software, San Diego, CA, USA) [23] was used for statistical data analysis. Ordinary 1-way ANOVA followed by Tukey’s multiple comparisons test was used for multiple comparisons. Student’s two-tailed t-test was used for pairwise comparisons. Data are presented as the mean ±SEM, unless otherwise indicated. In all figures and figure legends, ‘‘n’’ represents the number of samples or animals utilized in the indicated experiments. A *p*-value of less than 0.05 was considered significant. Ns, not significant; * *p* < 0.05; ** *p* < 0.01; *** *p* < 0.001.

## 3. Results

The study of cells in i.p. models has traditionally relied on terminal peritoneal lavages, in which mice are euthanized and then the peritoneal cavity is flushed with saline buffer. We hypothesized that by applying the same concept in vivo, with modifications, we could isolate i.p. fluid and assess the aspirate for cellular composition without affecting mouse survival. The workflow for our method, termed in vivo intraperitoneal lavage (IVIPL), is shown in Figure 1A.

We first tested administering different volumes of 1X PBS i.p. into tumor-free NSG mice. At a volume of 100 uL, it was difficult to aspirate any i.p. fluid. Increasing the volume to 200 uL allowed the aspiration of single digit uL of i.p. fluid, and the volume of aspirate obtained continued to increase with larger volumes of PBS injected. By administering 500 uL PBS, 50 uL i.p. fluid was reliably aspirated, and 500 uL injections of PBS were used for subsequent IVIPLs. IVIPL did not negatively affect the health of the mice as determined by observation, and, in fact, i.p. saline buffer injection is commonly used to help hydrate mice [24]. The mice remain under isoflurane anesthesia throughout the IVIPL procedure to limit the suffering of the animals. IVIPL isolated tens to hundreds of thousands of cells that can then be used for downstream analysis, such as flow cytometry.

Preclinical i.p. tumor models are often used to study ovarian cancer (OC) and potential treatments. To validate the application of IVIPL in xenograft OC models, we inoculated NSG mice i.p. with firefly luciferase and enhanced GFP (FG)-engineered high-grade serous ovarian cancer (HGSOC) cell lines, OVCAR8FG cells (Figure 1) or OVCAR3FG cells (Appendix A). Furthermore, as OVCAR8 and OVCAR3 cells express mesothelin (MSLN) and MSLN-targeted CAR-T (MCAR-T) cells are being actively tested in clinical and preclinical studies [25], we also studied the treatment of tumor-bearing mice with MCAR-T cells. A third generation CAR construct was used for these studies (Appendix A).

IVIPL did not affect the growth of OVCAR8FG cancer nor cause discernable toxicity (Appendix A). Following the experiment shown in Figure 1B, OVCAR8FG tumor-bearing mice were treated with MCAR-T cells or PBS. MCAR-T cells significantly slowed tumor growth but were unable to clear OVCAR8FG tumors and all mice succumbed to disease (Figure 1C,D). As shown in Figure 1E-J, IVIPL allowed the recurrent monitoring of cells in the i.p. cavity of NSG mice. Monitoring of human CD45^+^ cells and GFP^+^ tumor cells using flow cytometry revealed phenotypic changes in the therapeutic cell and cancer cell populations that may impact antitumor efficacy. Within the human CD45^+^CD3^+^ population there was a decrease in the percentage of CAR-positive therapeutic cells, an upregulation of inhibitory receptors PD-1 and TIM-3, and a concomitant increase in PD-1^+^TIM-3^+^ double positive cells. OVCAR8FG cells upregulated PD-L1 in the MCAR-T cell-treated group, indicating a potential mechanism for CAR-T cell immunosuppression. Although NSG mice are highly immunodeficient, the mice retain innate myeloid immune cells, allowing the study of host myeloid cell recruitment. OVCAR8FG tumor-bearing mice displayed increased percentages of tumor-associated macrophages as well as heightened expression of CD206 within the macrophage population. Importantly, IVIPL samples obtained five days before terminal peritoneal lavage indicate that IVIPL isolates accurately represent the cellular composition of the peritoneal cavity (Figure 1E,F, Day 30 IVIPL compared to Day 35 Terminal Lavage). IVIPL also identified MCAR-T cells in an OVCAR3FG tumor model (Appendix A). While NSG models are critical for studying human therapeutic and cancer cells, syngeneic models facilitate the examination of complex tumor microenvironments and have been instrumental in the study of cancer immunosurveillance and in the development of groundbreaking cancer therapies, such as immune checkpoint inhibitors [26,27]. The murine i.p. ID8 ovarian cancer model is extensively used for investigating HGSOC [28]. B6 mice were injected i.p. with ID8 cells or PBS, and IVIPL was performed twenty days later to monitor i.p. cellular compositions (Figure 2A–C). By day twenty post tumor inoculation, tumor-bearing mice experienced significant increases in the presence of tumor-associated macrophages and trends toward (but not statistically significant) decreases in proportions of effector NK1.1^+^ cells and CD8^+^:CD4^+^ T cell ratios. Retro-orbital bleeding of tumor-bearing and tumor-free B6 mice confirmed that cell composition changes were particular to the i.p. tumor microenvironment (Figure 2B).

## 4. Discussion

Preclinical i.p. models of intra-abdominal diseases, such as endometriosis [29], bacterial and viral infection [30,31,32], and cancer, especially ovarian cancer [33], are extensively used to analyze disease progression and develop treatments. A current bottleneck in studying the evolution of i.p. microenvironments in mice is the requirement of terminal peritoneal lavage or invasive procedures for i.p. cellular analysis. We have developed and validated a minimally invasive method, termed IVIPL, to overcome this limitation and allow for the continual monitoring of cell composition in the peritoneal cavity.

Some of the most difficult-to-treat cancers, such as ovarian, colorectal, and stomach carcinomas, often present at late-stage disease with peritoneal carcinomatosis and ascites [34]. Although initial responses to surgery and chemoradiotherapy are prevalent, relapse rates are high and five-year survival rates remain dismal [35]. Furthermore, current immunotherapies have failed to alter the treatment landscape for most ovarian and other intra-abdominal cancers [36,37], highlighting the urgent need for novel treatments.

CAR-T cell therapies have displayed remarkable success in combating B cell malignancies in the clinic but have made minimal headway into the treatment of solid tumors. Critical factors influencing CAR-T efficacy include therapeutic cell persistence, tumor infiltration, and functionality in the tumor microenvironment, which are linked to CAR-T cell phenotypic traits such as memory and exhaustion status [38,39]. Noninvasive cell-tracking methods employing MRI, PET, and optical imaging technologies are instrumental for studying cells in vivo but are predominantly limited to cell persistence and migration insights [40]. The TME is a complex milieu with an abundance of immunosuppressive cell populations, ligands, soluble factors, and metabolic restrictions [41]. A better understanding of the interactions between cancer cells, CAR-T cells, other immune cells, and stromal cells may lead to more effective cellular and combination therapies. When studying liquid cancers, peripheral blood is routinely used to characterize phenotypic changes in cell populations over time [42,43]. With IVIPL, analogous studies can be performed in the development of i.p. administered cell products. Of ten publications reviewed that investigated CAR-T cell therapy in the context of preclinical i.p. models, four used terminal lavages to assess the cellular composition of the peritoneal cavity at one or two time points and six did not assess the cells in the preclinical model [44,45,46,47,48,49,50,51,52,53]. Here, we studied MCAR-T cells in an NSG i.p. OVCAR8 model of HGSOC and witnessed increased i.p. infiltration of tumor-associated macrophages, decreases in the percentage of CAR^+^ therapeutic cells, and upregulation of inhibitory receptors and ligands on the therapeutic and OVCAR8 cells. These findings indicate that multi-pronged approaches targeting immunosuppressive myeloid cells and inhibitory receptors/ligands and further CAR-T cell engineering (cytokine secretion, BiTE expression, etc.) may be necessary to combat i.p. ovarian cancer. We also performed IVIPL and retro-orbital bleeds on ID8 ovarian cancer-bearing immunocompetent mice. In agreement with the NSG model, the syngeneic ovarian cancer cells heavily recruited macrophages to the peritoneal cavity. Despite an initial influx of effector T and NK cells, the balance shifted in favor of T regulatory cells and tumor-associated macrophages, and tumor progression ensued. Similar to previous CAR-T cell studies, ID8 investigations tend to rely on terminal peritoneal lavage for cellular analysis [54,55,56].

Preclinical models are a cornerstone for cancer drug development but it is clear that preclinical models do not automatically reflect the complexity of the TME in humans, as well as the differences between patient histotypes, stages of disease, and responses to treatment [57]. Over the past decade, increasing attention has been given to the characterization of components of patient ascites and their role in the progression of ovarian cancer [15]. Given ascites occurs in late-stage ovarian cancer, we propose the isolation of peritoneal fluid prior to ascites formation, such as during the first surgery for treating ovarian cancer, and, when possible, continual monitoring of peritoneal fluid during the course of disease (i.e., before and after hyperthermic intraperitoneal chemotherapy). The simultaneous application of IVIPL in preclinical models and characterization of human patient peritoneal fluid samples can add potential translational value to IVIPL-enabled studies and allow for comparisons between patient and animal model peritoneal TME evolution and response to therapy.

Our labs specialize in developing cell therapies, particularly CAR-T and hematopoietic stem cell-engineered invariant natural killer T (CAR-HSC-iNKT) cell therapies, for cancer treatment [58,59]. While we validated IVIPL for application in i.p. ovarian cancer models using flow cytometry, we envision its usage in other i.p. studies, such as in the burgeoning focus on tissue resident macrophages [60,61,62], and its pairing with other technologies (RNA-seq, ATAC-seq, CyTOF) and cellular characterization methods.

## 5. Conclusions

In vivo intraperitoneal lavage (IVIPL) is a novel method that can be used for the minimally invasive monitoring of cells in the peritoneal cavity of mice. An understanding of the i.p. microenvironment evolution in real time has the potential to uncover insights into disease progression and engender a more thorough study of treatment candidates.

## Figures and Tables

**Figure 1 cancers-14-01775-f001:**
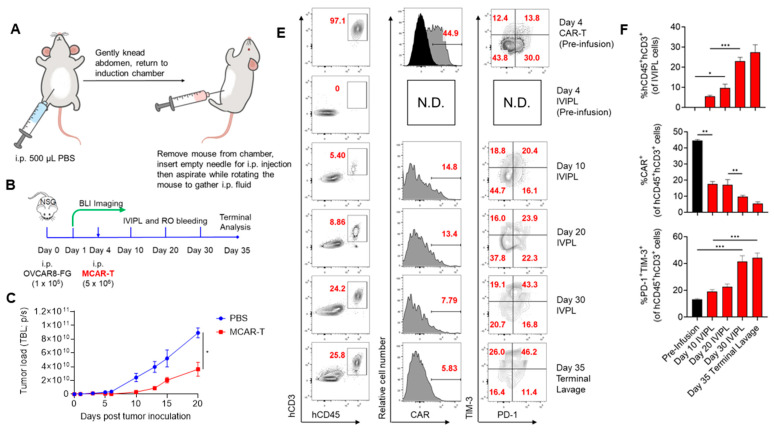
Design of in vivo intraperitoneal lavage (IVIPL) and its usage in studying CAR-T therapy. (**A**) Schematic representation of IVIPL. Mice are injected i.p. with 500 uL PBS and the abdomen is gently kneaded. Peritoneal fluid is then aspirated from the mice by inserting a needle while the mouse is held supine to avoid organs and rotating the mouse to allow fluid buildup in the lower abdomen. (**B**–**J**) Studying the in vivo antitumor efficacy of mesothelin-targeting CAR-T cells in an OVCAR8-FG human ovarian cancer xenograft NSG mouse model. (**B**) Experimental design to monitor CAR T cells using IVIPL in an ovarian cancer model. (**C**) Tumor growth curve (n = 4 per group) and representative bioluminescence images (**D**). (**E**) FACS plots showing the change in CAR-T cell phenotype over time, summarized in (**F**). hCD3^+^hCD45^+^ are gated on live IVIPL cells, CAR and PD1/TIM-3 plots gated on hCD45^+^CD3^+^ cells. Black bars indicate pre-infusion in vitro MCAR-T cells. (**G**) FACS plots showing the change in tumor PD-L1 expression between treatment groups on day 20, summarized in (**H**). Gated on GFP^+^ tumor cells. (**I**) FACS plots showing the change in tumor-associated macrophage accumulation and phenotype on day 20, summarized in (**J**). Left FACS plots gated on mCD45^+^ cells, right FACS plots gated on mCD45^+^mCD11b^+^F4/80^+^ cells. * *p* < 0.05; ** *p* < 0.01; *** *p* < 0.001.

**Figure 2 cancers-14-01775-f002:**
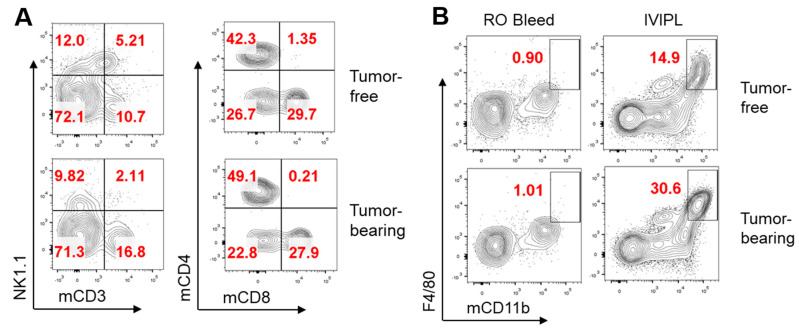
IVIPL can provide real-time insights into tumorigenesis in a syngeneic model. (**A**–**C**) B6 mice injected i.p. with 5 × 10^6^ ID8 cancer cells or PBS were assessed for peritoneal cavity and peripheral blood cell compositions 20 days later using IVIPL and RO bleeds, respectively (n = 2 per group). (**A**) FACS plots showing the presence of lymphocytes, left plots gated on mCD45^+^ cells, right plots gated on mCD45^+^mCD3^+^NK1.1^-^ (**B**) Representative FACS showing the presence of macrophages in peripheral blood and the peritoneal cavity in tumor-free and tumor-bearing mice. (**C**) Summary data of CD8^+^:CD4^+^ T cell ratios and CD8^+^:TAM ratios relative to the ratios in tumor-free mice. (**D**) Summary of MFI of CD206 on IVIPL isolated mCD45^+^mCD11b^+^F4/80^+^ cells. ns, not significant; * *p* < 0.05.

## Data Availability

The data presented in this study are available on request from the corresponding author.

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
