# Peer review of "Minimally Invasive Preclinical Monitoring of the Peritoneal Cavity Tumor Microenvironment"

_cancers, 2022, doi:10.3390/cancers14071775_

Round 1

Reviewer 1 Report

Submitted article represents a methodological study describing and evaluating a method termed in vivo intraperitoneal lavage (IVIPL), using murine models for syngeneic model of ovarian cancer, as well as a xenograft CER-T cell-treated ovarian cancer model.   Indeed, this method, unlike traditional intraperitoneal lavage, for which euthanizing mice is necessary, can offer o tool for monitoring of the cancer development- and treatment-related changes of a cellular landscape in the peritoneal cavity. The method was optimized, and a possibility to analyze the cellularities and phenotypes of particular immune cells populations was sufficiently documented. Importantly, no effects of repeated IVIPLs on the tumor growth or animal weights were observed.  IVIPL approach could beneficial, as it can allow monitoring of the i.p.  TME changes in time in a single animal. It can also help to reduce necessary numbers of animals used in experiments.

Specific comments:

  1. Can you discuss more in detail this approach from an ethical and animal welfare points of view, since IVIPL can lead to a reduction of animals used for experiments but, on the other hand, repeated treatments can be stressful and repeatedly painful for animals? What are the levels of suffering?
  2. What are the limits of the numbers and frequencies of the IVIPLs repeats?

Reviewer 2 Report

The manuscript titled “Minimally invasive preclinical monitoring of the peritoneal cavity tumor microenvironment” by Dunn ZS et al., is a proof-of-concept paper that aims to present a simple, novel method termed in vivo intraperitoneal lavage (IVIPL) for the minimally invasive monitoring of cells in the peritoneal cavity of mice. I agree with the authors that “in vivo” analysis of peritoneal microenvironment is crucial, especially in those tumors that develop intraperitoneally as ovarian cancer, and it give relevant and fundamental data both in term of prognostic classification and therapeutic perspectives.

At this regard it would be relevant to translate the preclinical model developed by the authors in clinical practice and assess the intraperitoneal lavage in ovarian cancer patients with a safe, feasible and valid method, both at diagnosis and during disease evolution.

Indeed, the tumor microenvironment is a dynamic assembly of different components (cancer, immune and stromal cells) and it changes in its composition in relation to disease evolution and specific treatments received by patients.

Overall, the topic is relevant, the model proposed very and the manuscript well written and exhaustive. I have some comments and some revisions to be addressed before acceptance.

Major comments:

  • I would focus the manuscript since the introduction on ovarian cancer, since the preclinical model was a model of ovarian cancer, and since ovarian cancer has its peculiarities in term of tumor microenvironment composition, therapeutic challenges and perspectives that are different from those of colorectal/gastrointestinal cancers.
  • In the Introduction I would suggest to the authors to briefly introduce the concept that tumor microenvironment composition (in ascites) may vary in different cancers and ovarian cancer subtypes, may change dynamically during cancer evolution, being influenced also by treatments and my significantly impact prognosis. At this regard the authors should also cite some relevant recent papers at this regard (for example Heath O, Berlato C, Maniati E, et al. Chemotherapy Induces Tumor-Associated Macrophages that Aid Adaptive Immune Responses in Ovarian Cancer. Cancer Immunol Res. 2021;9(6):665-681; Macciò A, Gramignano G, Cherchi MC, Tanca L, Melis L, Madeddu C. Role of M1-polarized tumor-associated macrophages in the prognosis of advanced ovarian cancer patients. Sci Rep. 2020;10(1):6096; and others) (after ref 7,8 in the Introduction for example)
  • Moreover, to better introduce CAR therapy and other immunotherapies cited in the Introduction, the authors should briefly state that the peritoneal tumor microenvironment is composed by several cells as cancer cells, stromal cells and immune cells that interact between them influencing tumor progression and immune escape and may represent specific targets of immunotherapy.
  • I agree that the model presented by authors is a very interesting method to study peritoneal tumor microenvironment in preclinical model and I agree that is could be superior to other methods used and cited by the authors in preclinical models. However, the authors should acknowledge that preclinical models and composition as well changes of tumor microenvironment in preclinical models do not automatically reflect the complexity of tumor microenvironment in humas as well as the difference between each hystotype, stage of disease and timing of collection in vivo in ovarian cancer patients, where peritoneal fluid could be collected and analyzed. Therefore, they could add the translational perspectives of their model and how it can help the in vivo studies in humans.
